# Free-Radical Photopolymerization for Curing Products for Refinish Coatings Market

**DOI:** 10.3390/polym14142856

**Published:** 2022-07-13

**Authors:** Aina Ribas-Massonis, Magalí Cicujano, Josep Duran, Emili Besalú, Albert Poater

**Affiliations:** 1Department of Chemistry, Institute of Computational Chemistry and Catalysis, University of Girona, c/Maria Aurèlia Capmany 69, 17003 Girona, Spain; ainaribasmassonis@gmail.com (A.R.-M.); josep.duran@udg.edu (J.D.); emili.besalu@udg.edu (E.B.); 2Roberlo S.A., Ctra. N-II, km 706,5, Riudellots de la Selva, 17457 Girona, Spain; mcicujano@roberlo.com

**Keywords:** coating, cross-linking, photopolymerization, curing, refinish

## Abstract

Even though there are many photocurable compositions that are cured by cationic photopolymerization mechanisms, UV curing generally consists of the formation of cross-linking covalent bonds between a resin and monomers via a photoinitiated free radical polymerization reaction, obtaining a three-dimensional polymer network. One of its many applications is in the refinish coatings market, where putties, primers and clear coats can be cured faster and more efficiently than with traditional curing. All these products contain the same essential components, which are resin, monomers and photoinitiators, the latter being the source of free radicals. They may also include additives used to achieve a certain consistency, but always taking into account the avoidance of damage to the UV curing—for example, by removing light from the innermost layers. Surface curing also has its challenges since it can be easily inhibited by oxygen, although this can be solved by adding scavengers such as amines or thiols, able to react with the otherwise inactive peroxy radicals and continue the propagation of the polymerization reaction. In this review article, we cover a broad analysis from the organic point of view to the industrial applications of this line of research, with a wide current and future range of uses.

## 1. Introduction

Putties, primers and clear coats are products designed for the refinish coatings market [1], with the purpose of protecting and decorating the bodywork (Figure 1), with special interest in the automotive market [2]. Putties are applied over the metal surfaces with the purpose of filling scratches. Primers protect the putty layer and promote adhesion to the base coat, which contains the visual property of color. Clear coats are transparent and provide surface properties including smoothness, gloss and weather [3] and light resistance [4,5]. Once layers are applied, the drying is another physical phenomenon that must be evaluated, and often involves two simultaneous processes, physical drying and chemical curing [6]. Physical drying involves the evaporation of the solvents present in the mixture.

Chemical curing involves a chemical reaction that results in the formation of a three-dimensional polymer matrix [7]. Traditionally, clear coats and primers cure through a non-catalyzed but stoichiometric reaction, and putties through a free radical polymerization reaction [8], all temperature-dependent processes.

In the field of polymer coating [9,10,11], there are different strategies that range from vegetable oil-based, environmentally friendly coatings [12,13,14] and epoxy acrylate resins of vegetable origin [15], or biodegradable lignin as a reactive raw material [16], to engineering the bio-nano interface using multifunctional coordinating polymer coatings [17]. However, a faster and greener alternative is ultraviolet (UV) curing, which consists of using UV light to initiate photochemical reactions. The reaction that takes place during UV curing is a free radical polymerization; the only difference between UV curing and the conventional thermal curing is the species that initiate the chain reaction [18]. The main advantages offered by UV curing are the possibility to define the start and end times of the initiation, the rapidity of curing as it only requires irradiation, which ranges from seconds to several minutes, and absence of a temperature requirement, translating into greater productivity and energy efficiency [19]. However, the main limitations are related to the curing of thick or pigmentated systems as they inhibit the penetration of light to the bottom [20]. This technique has already been implemented in many applications, such as wood finishing [21], biomedical engineering [22], dental materials [23,24] or adhesives [19,25,26,27], and more recently in the automotive refinish coatings market [20,28,29]. The applications of the coating are so extensive as to include even the keratin treatment of human hair, i.e., photoprotective coating of human hair [30], using its aqueous mixtures on a natural halloysite clay nanotube [31,32,33] that can be easily decorated [34,35,36].

Before considering the mechanistic insights of free radical photopolymerization, the advantages of the cationic UV curing process are enumerated: (a) in the polymerization, there is an absence of oxygen inhibition [37], which is the main problem of the radical photopolymerization, eliminating the need for an inert atmosphere [38,39]; (b) the absence of toxicity of the monomers used in cationic UV curing; and (c) lower residual stresses in the cured materials, which is translated into better adhesion properties on the substrates [40].

The structure of the review is outlined in [Fig polymers-14-02856-ch001], and the interest in linking photopolymerization with the term free radical is self-justified, beginning in the 1970s and experiencing steady progression to the hundred annual publications in 2019, with a subsequent expansion by an additional 50% over the last 2 years.

## 2. Free Radical Polymerization

During the cross-linked free radical polymerization reaction, oligomers and monomers functionalized with carbon–carbon double bonds react, forming single covalent bonds, resulting in a three-dimensional cross-linked polymer network (Figure 2). The sources of free radicals are photoinitiators [20,41,42].

### 2.1. Mechanism

The four basic steps of a free radical polymerization reaction are initiation, propagation, chain transfer and termination (Figure 1) [43,44]. Initiation creates free radical active centers, which then initiate the propagating chain; propagation adds consecutive monomer units to the growing polymer structure, since not every addition results in a cross-link. For instance, the addition of a monofunctional monomer leads only to the extension of the polymer chain, which is not a cross-link. Next, chain transfer ends a growing polymer and begins another (in Figure 1, H-R is any hydrogen donor, including a monomer, polymer or an additive), and termination destroys the active centers and ends polymer growth [43].

#### 2.1.1. Initiation

Initiation consists of two processes. During the first process, the initiator dissociates into two free radical species. In UV curing, the initiator decays upon irradiation with UV light [45], either by homolytic cleavage, hydrogen abstraction or electron transfer. Notably, in the last two cases, the donor radical is the initiating species.

In thermal curing, the initiator, usually a peroxide, decomposes thermally by cleavage of the oxygen–oxygen bond. This process can be accelerated by using a promoter; aside from a tertiary amine, as in Figure 2 [46], this can come from a range of different salts, including alkoxypyridinium [47], iodonium [48,49,50,51] or organoborate salts [52].

During the second process, either one or both of the radical species initiate a propagating chain by attacking a nearby monomer (Figure 3a), which can be part of an oligomer or a free monomer, and is incorporated into the polymer matrix [43,54,55].

#### 2.1.2. Propagation

Propagation involves the addition of monomers to the growing polymer (Figure 3b) [56]. Radicals react to form a covalent bond while generating a new radical, resulting in the formation of cross-links between oligomers [43,46].

#### 2.1.3. Chain Transfer

During chain transfer, an active center is transferred (Figure 3c). Propagating chains abstract a weakly bonded atom, usually a hydrogen or halogen, which cleaves homolytically. In consequence, a dead polymer chain and a new radical are generated, which can in turn continue propagating. The transfer can happen either inter- or intramolecularly [18].

#### 2.1.4. Termination

There are two types of termination. Termination by recombination involves the direct coupling of two propagating chains, which form one longer polymer. In termination, as shown in Figure 4, by disproportionation, one propagating chain abstracts a hydrogen atom from another propagating chain, yielding two stabilized polymer chains, of which one carries a double bond [18,43].

### 2.2. Inhibition by Oxygen

Photoinduced processes have gained attention and have led to the implementation of technological developments towards new polymeric products [57,58]. Despite the applications of polymerization photoinitiating systems in migration polymerization [59], one of the major disadvantages of free radical photopolymerization is its susceptibility to oxygen inhibition. Since oxygen is a biradical in its electronic ground state, it has the capacity to interact with a growing chain with a free radical active center, leading to a much less reactive peroxy radical. The latter is unable to break a C-H bond to undergo chain transfer (Figure 5). Consequently, this means that oxygen effectively acts as a chain terminator, reducing the rate of polymerization. In addition, in the vicinity of the surface, it is difficult to consume the oxygen faster than its diffusion, which can result in incomplete curing or tackiness on the surface layers [43].

Several methods can be used to minimize oxygen inhibition, such as using an inert gas, increasing the photoinitiator concentration or the light intensity to produce a larger number of active centers or adding oxygen scavengers to the mixture, such as amines or thiols. The peroxy radical can undergo chain transfer to thiols or amines since sulfur–hydrogen and N-H bonds are weaker, generating an active thiyl or alkylamino radical able to continue propagating (Figure 6) [43,60,61].

## 3. Constituents of a Free Radical UV Curing System

The essential constituents of a UV curing system are a resin [62], which is an oligomer whose backbone confers the properties to the final polymer; a monomer, which acts as a cross-linking agent and adjusts the viscosity of the mixture to an acceptable level for application; and a photoinitiator [63], which is responsible for the light absorbance and governs the curing depth and rate [20,41]. All of them participate in the cross-linked free radical polymerization reaction and are incorporated into the final polymer.

### 3.1. Resins

A resin is an oligomer, which is a chain formed by the union of monomer units, that will constitute the framework of the cured polymer network [55,64]. It cannot be considered a polymer because the latter is a macromolecule with a much larger number of monomer units [65], whereas these oligomers usually contain from 1 to 12 repetitive units [66]. They are usually formed through step-growth polymerization, a type of polymerization mechanism in which bifunctional or multifunctional monomers react to form first dimers, then trimers and, eventually, long-chain oligomers [67]. The type of monomer and their length, together with the cure extension, will determine the properties of the final polymer [38,68].

The main classes of UV-curable resins that can be polymerized by a radical mechanism are unsaturated and acrylate resins [20]. The most common backbone structure for unsaturated resins is polyesters, and for acrylate resins, they are polyurethanes, although other structures, such as polyesters, can also be used [69,70].

#### 3.1.1. Unsaturated Resins

Unsaturated resins periodically contain monomers with double bonds in their backbone, which will react during the free radical polymerization. They are generally polyesters, which means that they have ester linkages in their backbone chain, generated through condensation reactions between diols and unsaturated dicarboxylic acids, also called esterification reactions (Figure 7) [71]. More than one type of each reagent could be used, obtaining then an oligomer with over three different monomers. In any case, the unsaturations come from the structure of the diacid.

The Fischer esterification specifically refers to the acid-catalyzed reaction of carboxylic acids and alcohols. It is one of the methods that can be employed to synthesize polyester oligomers [72,73]. The alcohol from the diol nucleophilically attacks a protonated dicarboxylic acid, and after proton transfer, a water molecule is lost from the structure of the diacid. The resulting product is in an ester, which, since both reactants are difunctionalized, still contains an alcohol and a carboxylic acid group, able to further condensate and create a chain of esters (Figure 8).

#### 3.1.2. Acrylate Resins

Acrylate resins contain acrylate or methacrylate groups at their ends that will react during the free radical polymerization [74]. They are more efficient than unsaturated resins for UV curing. Urethane acrylates, which contain a polyurethane backbone, are the most common, but polyester or polyether backbones are also used (Figure 9) [43]. Given that all acrylates are derived from oil, which is a scarce material and also highly polluting, environmental pressure from climate change forces us to reduce the use of oil and/or look for alternative solutions, such as the conversion of renewable biomass into materials, polymers and composites [75]. The development and application of bio-based materials is therefore aimed at replacing commercial UV-curable acrylate resins. In detail, they are mainly epoxy, polyurethane and polyether acrylate oligomers. Resins have different functions depending on the chain structure. The evolution of the polymer chain structure has made polyester acrylate oligomers increasingly functional. Commercially, this has been replicated in the growing UV curing market. However, there are also drawbacks, caused by the fact that products of relatively low intensity are needed. Despite the disadvantage of high viscosity, polyester acrylates (PEA) lead to UV-curable resins with good hardness, high tear resistance and wear resistance, ozone resistance and polarity [76]. Looking for ways to reduce viscosity, and being part of the spectrum of non-petroleum-derived biodegradable polymers [77,78], poly(lactic acid) (PLA) and the use of biocurable UV coatings offer green advantages to the industry [79,80], PLA still has weaknesses although it is applied prematurely, and is far from the market. In particular, its resistance needs to be improved, especially at high temperatures. In order to influence improvements, in contrast to the linear polymers [81,82,83], the networking of PLA and the incorporation of star-shaped chains have been described, as well as modification by copolymerization with poly(ε-caprolactone) (PCL) to improve its hardness, also favoring the positive reduction of viscosity [84]. Notably, recently, maleimides have appeared to be competitors of acrylates in photopolymerization because they can operate without a photoinitiator and also because their polymerization rate is directly competitive with that of acrylates [85,86]. Nonetheless, polyurethanes are synthesized similarly to polyesters; however, the polyol reacts with a di- or triisocyanate instead of a diacid [87,88]. The polyol does not necessarily have to be a polyether; for instance, polyesters could also be used [89]. The synthesis is catalyzed by a tertiary amine, which, according to Farka’s mechanism, interacts with a proton source to form a complex that subsequently reacts with the isocyanate (Figure 10) [90].

In another type of curing, urethane cross-links are formed by the reaction of aliphatic polyisocyanate monomers with hydroxy-functionalized resins instead of monomers, obtaining a polyurethane polymer network [91], this process being not catalyzed but stoichiometric. Once synthesized, both polyester and polyurethane oligomers can react with a hydroxyl acrylate or methacrylate to form the acrylate terminations [92,93].

**Scheme 9 polymers-14-02856-sch009:**
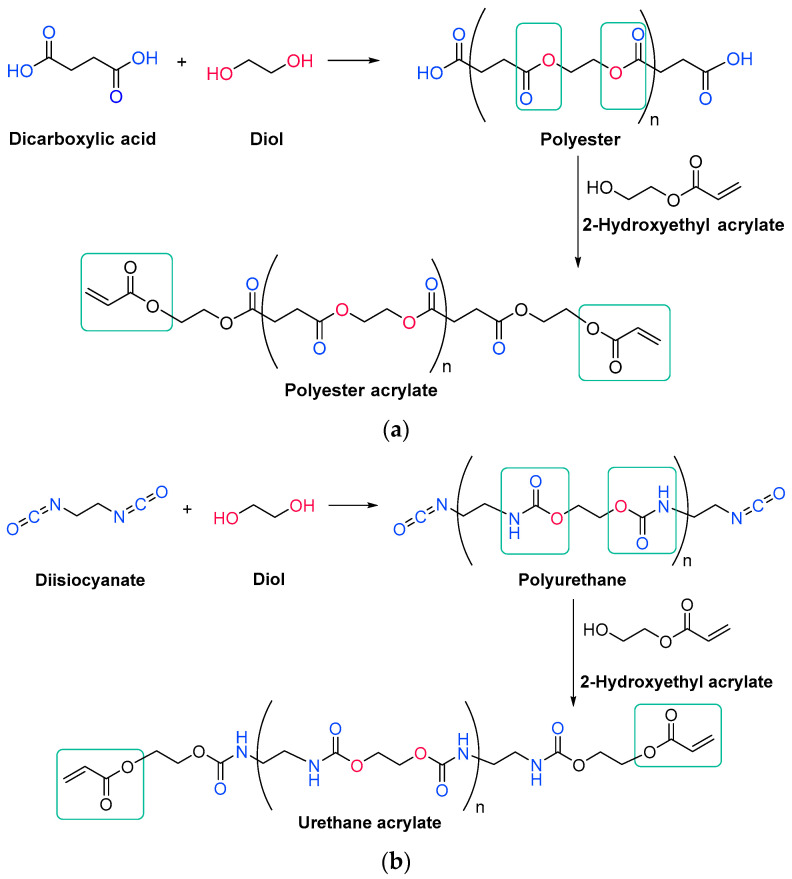
Generic synthesis of (**a**) a polyester acrylate oligomer and (**b**) a urethane acrylate oligomer [94].

**Scheme 10 polymers-14-02856-sch010:**
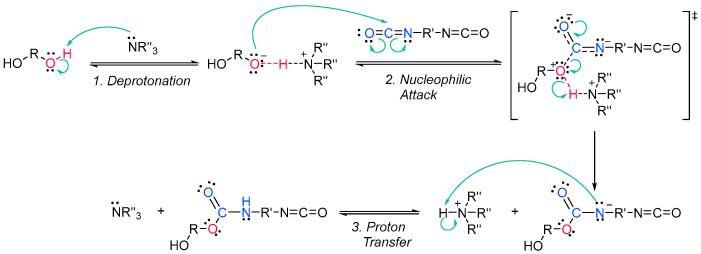
Farka’s carbamation mechanism [90].

The urethane acrylate oligomer can be divided into soft and hard sections. The soft segments are produced from the polyols, and the longer they are, the more flexible is the resin. The hard segments are produced from isocyanate and are immobile and stiff; moreover, they can form intermolecular hydrogen bonds between the hydrogen attached to the nitrogen and the carbonyl oxygen [89,95]. If a triisocyanate is employed instead of a diisocyanate, the resulting oligomer would have acrylates in the backbone too, which would be able to form cross-links during the free radical polymerization.

### 3.2. Monomers

Monofunctional monomers are used to lower the viscosity of the mixture and to add flexibility to the final polymer thanks to their cross-linking ability. In order to incorporate into the polymer, they must contain carbon–carbon double bonds [43]. Amongst the most common monofunctional monomers are unsaturated monomers, often used with unsaturated polyester resins, acrylate monomers [96], and usually paired with acrylate resins and thiol monomers, which can be added as curing accelerators [97,98].

The ratio of monomer will affect the rigidity of the final polymer, as the higher the concentration of monomer, the more probable the formation of consecutive bonds between monomers. In consequence, the cross-link chains will be longer and the polymer more flexible, which will help to reduce the stiffness and thus the risk of cracking in the case of a small collision. However, too much monomer could slow down the curing, since this means that more bonds would have to be created to form the polymer. In the following sections, the monomers are further detailed, divided into unsaturated, acrylate and thiol monomers.

#### 3.2.1. Unsaturated Monomers

The unsaturated monomer per excellence is styrene, a small molecule with little steric hindrance in comparison to the resin. Figure 11 depicts how styrene forms cross-linked chains between unsaturated oligomers.

#### 3.2.2. Acrylate Monomers

A great variety of acrylate monomers are available, so that the ideal monomer or combination of monomers can be chosen in order to adjust the flexibility of the final polymer (Figure 12).

The number of unsaturated groups will affect the flexibility of the final polymer. A higher number of unsaturations will form a higher number of cross-links and will cause the polymer to be more rigid. On the other hand, flexibility can be increased by using either long-chain linear monomers such as 1,6-hexanediol diacrylate, which will link oligomers while still allowing them to move, or bulky monomers with high steric effects, such as isobornyl acrylate, which will hinder the formation of cross-links near them [97].

A special type of acrylate monomer is phosphates (Figure 13a). They act as adherence promotors to metal surfaces, making them useful in certain applications [99]. Phosphate anions can replace hydroxy anions on metal oxide surfaces, so that the phosphate is adsorbed into the metal surface, making its removal difficult since chemical bonds have been formed [100].

#### 3.2.3. Thiol Monomers

Thiol monomers combined with ene monomers can form thiol-ene systems, which are suitable UV-curable resins (Figure 13b). However, monomers are more volatile than oligomers. Hence, thiols are rarely used as monomers due to their unpleasant odor, but can be used as oligomers, which are less volatile [43].

Their main advantage is that little to no photoinitiator is required in order to polymerize since thiols can function both as monomers and photoinitiators [101]. When exposed to UV light, they produce a thiyl and a hydrogen radical pair through sulfur–hydrogen bond cleavage [43]. However, this process is not as efficient as with an initiator; for this reason, the initiating species is often generated from the hydrogen abstraction reaction between a photoinitiator and the thiol (Figure 14) [102,103]. The resulting thiyl radical adds to a double bond of a monomer and, from here, the rest of the mechanism proceeds in the same way as in Figure 1. Given that most of the thiols used are polyfunctionalized, they act as powerful cross-linking agents [56,104].

Another advantage offered by thiols is that they are less sensitive to oxygen inhibition than acrylates [98]. For these reasons, thiols can be used as comonomers to accelerate the initiation of the free radical polymerization and to achieve better curing on the surface, but do not necessarily substitute the respective unsaturated or acrylate monomers [43].

### 3.3. Photoinitiators

Photoinitiators are able to convert light energy into chemical energy in the form of a reactive species, which can be radicals or cations, leading to the initiation of the polymerization chain. In the case of free radical polymerization reactions, radical photoinitiators are used. They are considered essential components of UV curing systems because most of the commonly used monomers are not able to generate free radicals upon exposure to UV light [20].

Many free radical photoinitiators are based on the benzoyl chromophore and can therefore undergo Norrish-type reactions. Norrish Type I photoinitiators generate the free radicals via an α-cleavage. On the other hand, Norrish Type II photoinitiators form free radicals through hydrogen abstraction, which can be either intramolecular or from a co-initiator. However, other mechanisms can be followed to generate the radical active species depending on whether the photoinitiator system is unimolecular or bimolecular [43]. It should be stated also that the original Norrish Type II cleavage reaction worked by intramolecular hydrogen abstraction followed by a C-C bond cleavage that did not lead to any radical species capable of the initiation of free radical polymerization.

#### 3.3.1. Unimolecular Photoinitiators

Photoinitiator systems termed unimolecular involve only one molecular species to generate the radical active species through homolytic cleavage. These photoinitiators are typically acetophenone derivatives, benzoin ethers, amino ketones or phosphine oxide derivatives. In most cases, the cleavage may occur in the α-position to the carbonyl group (Norrish Type I), but it can occur at the α-position in the presence weak bonds such as carbon–halogen, carbon–nitrogen, carbon–oxygen or carbon–sulfur next to the carbonyl moiety (Figure 15). One of the products of α-cleavage is always a benzoyl radical, while in β-cleavage, it is always a phenacyl radical. The other radical formed will depend on the structure of the initial photoinitiator, and it will not always be active—it could often be disproportionate or recombine [43].

Other types of unimolecular photoinitiators are those that can form biradicals through intramolecular hydrogen abstraction (Norrish Type II). This occurs in molecules with a hydrogen atom in the α-position, able to undergo an intramolecular [1,5]-hydrogen shift. The resulting ketyl radical will most likely terminate by coupling with another free radical species, while the other radical will initiate polymerization [43,46].

Upon the absorption of light with a specific frequency, photoinitiators are promoted from the ground electronic state to an excited singlet state, from where they can undergo inter-system crossing to a triplet state of comparable energy (Figure 3). It is from the triplet state that the molecule will cleave, generating the radical species [20,107]. However, Figure 3, supplemented with the cleavage reaction of triplets only, could be somewhat incomplete since the unimolecular photoinitiators may cleave either from the singlet or triplet state, depending on the photoinitiator structure, but the cleavage of singlets is more likely due to their higher energy. Bimolecular photoinitiators work usually from the triplet state, because the lifetime of most of singlet states is too short to enable bimolecular reactions.

#### 3.3.2. Bimolecular Photoinitiators

Photoinitiator systems termed bimolecular involve a photoinitiator that absorbs light, and a co-initiator that serves as a hydrogen or electron donor. In both cases, the formation of radicals takes place when the photoinitiator is either in the singlet or triplet excited states. These photoinitiators are typically benzophenone derivatives, thioxanthones, camphorquinones, benzyls or ketocoumarins [43].

In initiation by hydrogen abstraction, the co-initiator is usually an ether or an alcohol with an α-hydrogen. The resulting ether or alcohol radical will be the only initiating species. On the other hand, in photoinitiation by electron transfer, the co-initiator is typically an amine, and it forms an excited-state complex with the photoinitiator, from where electron transfer occurs. It is immediately followed by the proton transfer of an α-hydrogen from the amine, resulting in an active amine radical capable of initiating polymerization. The fact that an amine is present in the system helps to neutralize oxygen inhibition. In both cases, a ketyl radical is formed, which only participates in termination (Figure 16) [43,46,108].

In addition, there are other bimolecular photoinitiators, not described here, which are among the most common components used in photoinitiating systems in the last decade, including, among others, polymethine dyes [52,109], squaric acid derivatives [110,111,112] or BODIPY dyes [112,113].

### 3.4. UV Light

UV light could be considered the fourth essential component of UV curing. In physics, the term ‘light’ refers to electromagnetic radiation of any wavelength. The spectrum of electromagnetic radiation can be organized by decreasing wavelength and thus increasing energy into radio waves, microwaves, IR radiation, visible light, UV radiation, X-rays and gamma rays (Figure 4). Radiation within the UV spectrum can be further divided by wavelength into UVA (315–400 nm), UVB (280–315 nm) and UVC (100–280 nm) [114]. The sun emits mainly visible light and infrared radiation, but it also emits some UV radiation. Of the UV light that reaches the Earth’s surface, more than 95% is UVA, with a small remainder of UVB and almost no UVC [115].

Each type of radiation will interact differently with matter depending on how energetic it is. Microwaves only cause changes in the rotational states of atoms and molecules, IR radiation can also trigger vibrational transitions [116], visible and UV light are energetic enough to modify the electronic structure by exciting outer-shell electrons, and X-rays can excite inner-shell electrons (Figure 5).

The fact that UV and visible light can excite valence shell electrons means that it can trigger chemical reactions such as free radical photopolymerization. In organic molecules containing σ, π and n electrons, the absorption of UV–vis radiation is restricted to those molecules that contain chromophore functional groups with valence electrons of low excitation energy, such as photoinitiators. The electronic transitions that may occur in these systems are depicted in Figure 6. However, among the outlined transitions, only n → π* and π → π*, the two lowest in energy, are available in the UV–vis spectrum [117,118].

For the UV curing to be viable, the absorption spectrum of the photoinitiator must overlap with the emission spectrum of the light source [43]. The most common types of UV lamps are mercury and Light Emitting Diode (LED) lamps. The irradiation wavelength for mercury lamps ranges from 185 to 650 nm; however, they are being substituted by the less hazardous LED lamps, which irradiate at a much narrower range of 390–400 nm, the least energetic UV radiation. However, the recent developments in LED technology, emitting at 365–370 nm, have allowed the design of novel, powerful and efficient light sources that lead to the free radical and cationic photopolymerization of monomers [119], up to the synthesis of interpenetrating polymer networks (IPNs) [120]. Since it emits at the edge between UV and visible light, systems that can cure with LED lamps might also cure with natural light, although slowly, as the irradiance, a determining factor in the curing rate, will be lower. The use of LED lamps limits the number of available photoinitiators.

Figure 7 depicts the absorption spectrum of some unimolecular and bimolecular photoinitiators [121]. It can be seen how photoinitiators A and C absorb at the UV LED wavelength range, while photoinitiators B and C would not be activated with an LED lamp.

The overlap between the initiator and light source must preferably not coincide with the absorption peaks of other components in the photopolymerization [122], such as monomers or pigments. In systems where there is overlap, higher light intensities and photoinitiator concentrations are often used. Even so, achieving a fast, deep and thorough cure in thick, pigmented coatings by UV radiation remains one of the greatest challenges in photocuring [123,124,125]. Light scattering by the pigment particles will prevent the penetration of photons in the deep-lying layers. A possible solution would be the use of photosensitizers, molecules able to absorb at longer wavelengths and to transfer their excitation to a photoinitiator [20].

In a clear formulation, UV radiation is absorbed mainly by the photoinitiator, so that the cure depth is directly controlled by the photoinitiator. Photoinitiated curing follows a surface to depth gradient because of the limited penetration of light; for this reason, there is an optimum initiator concentration for the efficient curing of thick samples. As the initiator concentration is increased, the initiation rate at the top of the film is increased, but at the same time, the initiation rate at the bottom is decreased because the photoinitiator molecules at the top absorb most of the UV light, impeding its penetration to the bottom [20].

## 4. Additives

Apart from the essential constituents of a UV curing system [126], other ingredients are added to the formulations to enhance their properties. However, these additives do not participate in the polymerization reaction and therefore are not covalently linked to the final polymer [127]. The type and quantity of additives will determine whether the resulting formulation will be a putty, a primer or a clear coat.

### 4.1. Mineral Fillers

Several mineral fillers can be used for each putty or primer in the form of white powders. They are called fillers because, traditionally, they have been used to reduce the cost of the formulation. The most important mineral filler is talc, a hydrated magnesium silicate with chemical formula Mg_3_Si_4_O_10_(OH)_2_ [128]. Since talc is the softest filler, it allows for a smooth sanding down; however, the main drawback of talcum powders is their high cost, which needs to be compensated with other fillers such as carbonates or sulfates. Among the most used are calcium carbonate (CaCO_3_), calcium sulfate (CaSO_4_) and barium sulphate or baryte (BaSO_4_).

### 4.2. Thinners/Thickeners

Several thickening agents can be used to increase the viscosity of the product without changing any other properties. The most common one is aerosil, also known as pyrogenic silica. Aerosils are agglomerates of silica particles that have extremely low bulk density [129]. On the other hand, thinners are also used to lower the viscosity—for instance, by means of reactive diluents/small monomers of low viscosity [130].

### 4.3. Microspheres

Microspheres are microscopic spheres made of glass, which are empty on the inside, and they are available with different diameter sizes [131]. They are exclusively used for putties since their purpose is to increase the volume of the final mixture in order to obtain a dough texture.

### 4.4. Pigments

The purpose of pigments is to provide good coverage to the putty or primer so that, when they are applied, light does not pass through [132]. This allows them to disguise where the reparation has been made once the paint coat is applied. They are mostly inorganic; the white ones provide very good coverage and can be used in large amounts as fillers, while the black ones can give different tones of gray to the product. On the other hand, organic pigments give a stronger color tone while not increasing the coverage significantly.

### 4.5. Solvents

Solvents are present in primers and clear coats to lower their viscosity for a smooth application with the spray gun, but not in putties, which do not undergo physical drying.

## 5. Conclusions

The aim of this review is to develop interest in new ranges of UV curing products for the refinish coatings market, especially for the automotive market, aside from the known cationic photopolymerization, as an alternative mode of curing. UV curing may include putties, primers, putty–primers and/or clear coats. The objective of developing UV curing products is to drastically reduce curing times, which translates into greater productivity but also higher energy efficiency, as this type of curing does not require an external source of temperature to proceed faster. Although there are still unresolved challenges, particularly the penetration of light to the bottom of the material, the future will be characterized by industrial applications that require UV curing, rather than by what is developed in basic research; in fact, to a greater or lesser extent, the field will be required to adapt to the market demands—in particular, the automotive market.

## Data Availability

Not applicable.

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
