# Peer review of "Free-Radical Photopolymerization for Curing Products for Refinish Coatings Market"

_polymers, 2022, doi:10.3390/polym14142856_

Round 1

Reviewer 1 Report

Dear Authors,

This review article is focused on the basic information about photopolymerization process and its application in production of refinish coatings materials.

In my opinion, more information about photoinitiators should be presented. There are many bimolecular photoinitiators not described here.

For example, what about polymethine dyes, squaric acid derivatives or BODIPY dyes, that are the most common components used in photoinitiating systems in the last decade.

Authors, said, that an amine is typically a co-initiator acting via electron-transfer process. I don’t agree with this. There are known other compounds playing a role of co-initiators, for example: alkoxypyridinium salts, iodonium salts, organoborate salts, …

Author Response

This review article is focused on the basic information about photopolymerization process and its application in production of refinish coatings materials. In my opinion, more information about photoinitiators should be presented. There are many bimolecular photoinitiators not described here. For example, what about polymethine dyes, squaric acid derivatives or BODIPY dyes, that are the most common components used in photoinitiating systems in the last decade.

OUR ANSWER: We apologize for this omission, at least point out other types of bimolecular photoinitiators, mainly during the last 5 years. We have added a new piece of text with the new references 103-108, by Jurek, Kowalczyk, Dumur, Kabatc and coworkers in refs. 109-113.

Authors, said, that an amine is typically a co-initiator acting via electron-transfer process. I don’t agree with this. There are known other compounds playing a role of co-initiators, for example: alkoxypyridinium salts, iodonium salts, organoborate salts, …

OUR ANSWER: We must agree that the word “typically” was not absolutely true, and the coinitiator nature can be different, and accordingly we have modified the text, including other types from Paczkowski, Jurek, Versace, Ortyl, Kabatc, Kowalczyk and coworkers in refs. 47-52.

Reviewer 2 Report

In this work, the authors discuss recent advances in free-radical photopolymerization of curing products for refinish coatings. The article presents a timely overview of this topic, elaborating different components and mechanisms contributing to the polymerization.

Overall, the details of each component or particular question are relatively well explained but a clear general view is missing, the presentation is more a compilation of different cases rather than a unified view. My main criticism is that a more unified/global presentation of the subject is needed, instead of a list of particular cases. Some sort of table, graphical summary, cartoon, scheme may help to have a global view of the subject. I miss a more general view of the problem and the mechanisms involved (a sort of roadmap) before a detailed discussion of each particular case. This must be addressed in the revised version.

I have also specific comments about particular issues:

The aim of this review is to develop interest in new ranges of UV curing products for the refinish coatings market, apart from the known cationic photopolymerization as an alternative mode of curing. So the author needs to highlight the advantages and benefits over the previous cationic photopolymerization, in the discussion.

Future direction and challenges of this subject is not clear, like how to tackle the issue of penetration of light into the bottom.

Author Response

In this work, the authors discuss recent advances in free-radical photopolymerization of curing products for refinish coatings. The article presents a timely overview of this topic, elaborating different components and mechanisms contributing to the polymerization.

OUR ANSWER: We thank the Reviewer for such a short and clear definition of our Review.

Overall, the details of each component or particular question are relatively well explained but a clear general view is missing, the presentation is more a compilation of different cases rather than a unified view. My main criticism is that a more unified/global presentation of the subject is needed, instead of a list of particular cases. Some sort of table, graphical summary, cartoon, scheme may help to have a global view of the subject. I miss a more general view of the problem and the mechanisms involved (a sort of roadmap) before a detailed discussion of each particular case. This must be addressed in the revised version.

OUR ANSWER: We thank the Reviewer for the advice, and for a better understanding, we have included new Chart 1, that summarizes the content included in the review.

I have also specific comments about particular issues:

The aim of this review is to develop interest in new ranges of UV curing products for the refinish coatings market, apart from the known cationic photopolymerization as an alternative mode of curing. So the author needs to highlight the advantages and benefits over the previous cationic photopolymerization, in the discussion.

OUR ANSWER: Apart from the separated comments, before entering into details of the mechanism we have pointed out the main advantages, as well as we have added new references 36-40 of Chiappone, Lalevée, Crivello and coworkers. In addition, last advances in photoinitiators have been included in new references 109-113.

Future direction and challenges of this subject is not clear, like how to tackle the issue of penetration of light into the bottom.

OUR ANSWER: We have added a brief comment in the conclusions claiming that future will be led by the interest in the field from industry, and then the basic research and these applications required will have to match, specially the difficulty of the penetration of light to the bottom.

Reviewer 3 Report

1. In the introduction section, the author emphasize the significance areas and future view of this work.

2. A Statistical report about the published papers during the two last decades according to the web of science or engineering village or etc. be added into the introduction section.

3. A comparative explanation or tables about the various monomers be added in section 3.2.

4. Although, this review papers describe or introduce the various photopolymerization method, but, introducing the various researches and their results can increase the rate of this review paper. Also, adding some tables can be helpful.

5. A separate section with the title of future trend be added into the manuscript.

Author Response

  1. In the introduction section, the author emphasize the significance areas and future view of this work.

OUR ANSWER: As stated in point 5, we have added a discussion on the future view, and the main challenge to be solved in this UV curing field. In addition, new references 36-40 of Chiappone, Lalevée, Crivello and coworkers have been added to compare advantages of the methods discussed here, as well as new references 109-113 on the last advances in photoinitiators.

  1. A Statistical report about the published papers during the two last decades according to the web of science or engineering village or etc. be added into the introduction section.

OUR ANSWER: We have added a new piece of text with a simple comparison: “The interest in linking photopolymerization with the term free radical is self-justified with the beginning in the 70s and a steady progression to the hundred annual publications in 2019, with a subsequent explosion of an additional 50% over the last 2 years..”

  1. A comparative explanation or tables about the various monomers be added in section 3.2.

OUR ANSWER: We have added a comment to clarify which types of monomers are used, enumerating them.

  1. Although, this review papers describe or introduce the various photopolymerization method, but, introducing the various researches and their results can increase the rate of this review paper. Also, adding some tables can be helpful.

OUR ANSWER: For a better understanding we have included new Chart 1, with the content of the review. If this schematic way by means of summary of content is considered not enough for sure we will add graphics with arrows for example.

  1. A separate section with the title of future trend be added into the manuscript.

OUR ANSWER: We have added a brief comment in the conclusions claiming that future will be led by the interest in the field from industry, and then the basic research and these applications required will have to match.